# Reduction of oxidative stress on DNA and RNA in obese patients after Roux-en-Y gastric bypass surgery—An observational cohort study of changes in urinary markers

**Elin Rebecka Carlsson**[1,2]*, **Mogens Fenger**[1], **Trine Henriksen**[3], **Laura Kofoed Kjaer**[3], **Dorte Worm**[4], **Dorte Lindqvist Hansen**[5], **Sten Madsbad**[6], **Henrik Enghusen Poulsen**[3]*

1 Department of Clinical Biochemistry, Copenhagen University Hospital Hvidovre, Hvidovre, Denmark, 2 Department of Clinical Biochemistry, Nordsjaellands Hospital, University of Copenhagen, Hilleroed, Denmark, 3 Department of Clinical Pharmacology, Bispebjerg Frederiksberg Hospital, Copenhagen University Hospital, Copenhagen, Denmark, 4 Department of Medicine, Amager hospital, Copenhagen, Denmark, 5 Steno Diabetes Center Copenhagen, Gentofte, Denmark, 6 Department of Endocrinology, Copenhagen University Hospital Hvidovre, Hvidovre, Denmark

* elin.rebecka.carlsson@regionh.dk (ERC); henrik.enghusen.poulsen.01@regionh.dk (HEP)

**Data Availability Statement:** Data were extracted from hospital repositories and contain potentially

## Abstract

Increased oxidative stress in obesity and diabetes is associated with morbidity and mortality risks. Levels of oxidative damage to DNA and RNA can be estimated through measurement of 8-oxo-7,8-dihydro-2′-deoxyguanosine (8-oxodG) and 8-oxo-7,8-dihydroguanosine (8-oxoGuo) in urine. Both markers have been associated with type 2 diabetes, where especially 8-oxoGuo is prognostic for mortality risk. We hypothesized that Roux-en-Y gastric bypass (RYGB) surgery that has considerable effects on bodyweight, hyperglycemia and mortality, might be working through mechanisms that reduce oxidative stress, thereby reducing levels of the urinary markers. We used liquid chromatography coupled with tandem mass spectrometry to analyze the content of 8-oxodG and 8-oxoGuo in urinary samples from 356 obese patients treated with the RYGB-procedure. Mean age (SD) was 44.2 (9.6) years, BMI was 42.1 (5.6) kg/m$^2$. Ninety-six (27%) of the patients had type 2 diabetes. Excretion levels of each marker before and after surgery were compared as estimates of the total 24-hour excretion, using a model based on glomerular filtration rate (calculated from cystatin C, age, height and weight), plasma- and urinary creatinine. The excretion of 8-oxodG increased in the first months after RYGB. For 8-oxoGuo, a gradual decrease was seen. Two years after RYGB and a mean weight loss of 35 kg, decreased hyperglycemia and insulin resistance, excretion levels of both markers were reduced by approximately 12% ($P < 0.001$). For both markers, mean excretion levels were about 30% lower in the female subgroup ($P < 0.0001$). Also, in this subgroup, excretion of 8-oxodG was significantly lower in patients with than without diabetes. We conclude, that oxidative damage to nucleic acids, reflected in the excretion of 8-oxodG and 8-oxoGuo, had decreased significantly two years after RYGB—indicating that reduced oxidative stress could be contributing to the many long-term benefits of RYGB-surgery in obesity and type 2 diabetes.

identifying or sensitive patient information that, due to local regulations and current legislation, cannot be shared publicly. Data requests may be sent either directly to the authors or to the department of Clinical Biochemistry, Copenhagen University Hospital Hvidovre (contact via kliniskbiokemi. hvidovrehospital@regionh.dk or at +45 38 62 11 00).

**Funding:** The collection of samples to the biobank used in this study was partially funded by The Ministry of Higher Education and Science (the UNIK project). The authors received no specific funding for their work related to this study.

**Competing interests:** The authors have declared that no competing interests exist.

# Introduction

Obesity increases the risk of progression to metabolic syndrome, type 2 diabetes, cardiovascular- and liver disease, as well as certain cancers and is generally associated with an increased all-cause mortality rate [1, 2]. A combination of increased inflammation and oxidative stress, induced by obesity, is suggested to be causative [2–4]. Oxidative stress, broadly defined as an imbalance in the body's naturally occurring oxidation- and reduction processes which leads to a net increase in concentrations of highly reactive oxygen species (ROS) [5], further seems to accelerate the development of micro- and macrovascular complications in patients with diabetes [6].

Oxidative stress can damage important cell-structures like biological membranes, proteins, lipids, DNA and RNA. Under normal conditions, damage caused by ROS are kept at a minimum through the acts of the body's own defense system [5]. The products of the repair-mechanisms controlling oxidative damage to nucleic acids can be assessed in urine as the markers 8-oxo-7,8-dihydro-2´-deoxyguanosine (8-oxodG) [7] and 8-oxo-7,8-dihydroguanosine (8-oxoGuo) [8], respectively. 8-oxodG is the result of oxidation of the DNA guanine moiety and 8-oxoGuo is the result of oxidation of the RNA guanine moiety [7]; they are produced when 8-hydroxylation of guanine (the most vulnerable of nucleobases [5]) is repaired or degraded in DNA or RNA, respectively [8], and technically they can be differentiated due to the difference in the ribose and the deoxyribose part of the molecule [8]. Guanine oxidation is known to destabilize the guanine-quadruplexes (so called G4's) established in the single-strand stretches of promoters (thereby affecting gene transcription) and telomeres [9]. Up to half of the oxidized and damaged DNA is thought to originate from the telomeres [10], suggesting that oxidative stress has an important impact on the ageing process. The exact oxidative mechanism is not clear as several pathways are possible, including hydroxyl radical oxidation [11].

Both 8-oxodG and 8-oxoGuo have shown to be associated with metabolic diseases, including type 2 diabetes [8, 12, 13] and excretion of both 8-oxodG [14] and 8-oxoGuo [15] has been reported increased in obese populations compared to controls. In type 2 diabetes, 8-oxoGuo excretion is higher in patients with than without complications [16] and is prognostic, perhaps even predictive, of mortality-risk and risk of death from macrovascular complications [17, 18].

Little is known about the effect of weight loss on oxidative damage on DNA and RNA. In mice, a recent study found a reduction in DNA damage (assessed with the comet assay) after weight loss in several inner organs [19]. In humans, the same method has been used to show a reduction in lymphocyte DNA damage after weight loss induced by metabolic surgery, in 56 patients twelve months post-operative [20]. The same research group later confirmed its results in vivo, with micronucleus assessment in a similar patient cohort (the type of metabolic surgery performed was not specified in either of the manuscripts) [21]. Nevertheless, these studies together with a report of decreased levels of urinary 8-oxodG, twelve months after laparoscopic sleeve gastrectomy in twenty-one morbidly obese patients [14], indicate that weight loss and/or metabolic surgery might be effective in reducing DNA oxidation. In the latter study, urinary 8-oxodG levels fell gradually to around half the rate twelve months after sleeve gastrectomy, a level that was close to the control group of hundred healthy, non-obese volunteers [14]. The RNA marker 8-oxoGuo was not measured.

Roux-en-Y Gastric bypass (RYGB) surgery has a high success-rate in reducing weight, normalizing dyslipidemia and lowering hyperglycemia in both the short- and the long-term [22, 23]. Furthermore, RYGB was proven able to reduce mortality in obese patients with and without diabetes [24]. It is still unknown, if some of these effects are facilitated or sustained through a reduction in oxidative stress. Most of the studies investigating one or several different markers of oxidative stress and antioxidant defense after RYGB in obese patients suggest an

improvement of oxidative stress after the surgery, although results are partly conflicting—for instance some of the studied markers (glutathione, superoxide dismutase and catalase) have been reported to be increased after RYGB in some of the studies and to be decreased in other studies [25–31]. How levels of oxidative DNA and RNA damage, reflected in the urinary markers 8-oxodG and 8-oxoGuo, respectively, are affected by RYGB, has however not been reported, except for a study on obese Zucker rats [32].

This paper reports our findings from a study in a large cohort of 356 patients, where we aimed to examine if a reduction in oxidative stress level (reflected in the levels of the urinary markers 8-oxodG and 8-oxoGuo) could be detected within the first two years after RYGB-surgery. We also aimed to find out if there were differences in the levels of the urinary markers between patient subgroups based on diabetes status and also, if levels differed before or after surgery depending on the outcome after surgery on the diabetic condition. Finally, we aimed to investigate whether there were associations between changes in the oxidative stress markers and other beneficial changes (for example weight loss and improvements in glucose- and lipid metabolism) after RYGB.

We hypothesized *a priori*, that because the urinary markers 8-oxodG and 8-oxoGuo represent an indirect measure of oxidative damage to nucleic acids that is increased in obesity and type 2 diabetes, one would expect a decrease in levels of the two urinary markers after RYGB-surgery, along with the reduction in weight, hyperglycemia and insulin resistance. We also hypothesized that levels of the urinary markers would be higher in patients with type 2 diabetes and possibly, correlated to weight change and/or markers of glucose- and lipid metabolism.

## Materials and methods

### Research population

The research population is a cohort of patients, mainly of Caucasian ethnicity, who had surgery for obesity with the RYGB procedure between November 2010 and September 2013 and attended pre-examinations and postoperative medical follow-up at Copenhagen University Hospital Hvidovre, Denmark [33]. The criteria for inclusion were a BMI above 35 together with one or several comorbidities related to obesity (type 2 diabetes, resistant hypertension, sleep apnea, infertility in women with polycystic ovary syndrome, lower extremity arthrosis), or a BMI above 50 without complications. Minimum age was 25 years. Exclusion criteria were, apart from general contraindications for surgery and/or anesthesia, severe psychiatric disorder, eating disorder or substance abuse. All patients were instructed to take vitamin and mineral supplementations according to international nutritional recommendations and common practice [34]. Unable- or unwillingness to follow the nutritional recommendations was an exclusion-criteria.

As illustrated in S1 Fig, we have included patients who between October 2010 and September 2015, before as well as on minimum one occasion after their RYGB-operation, had delivered urine and blood samples to a research biobank. A total of 356 out of 786 RYGB-operated patients were included, for whom collective data from 1) the urinary analysis, 2) measurement of cystatin C in a matching blood sample and 3) timely recorded data on height, weight and routinely measured plasma creatinine enabled calculation of pre- and postoperative 24-hour 8-oxoGuo and 8-oxodG excretion, as described in the section below. Altogether, 24-hour estimates of 8-oxoGuo and/or 8-oxodG excretion could be calculated in 1254 urine samples, where 356 samples (from 356 patients) were collected before surgery, and 269, 232, 229 and 168 samples were collected at planned intervals of three, six, twelve and twenty-four months after RYGB, respectively, as shown in Table 1. For sixty-nine of the patients, a 24-hour excretion estimate could be calculated for 8-oxoGuo at all four timepoints after RYGB, and for 118,

**Table 1. Adherence of the 356 included patients to post-operative follow up (example for 8-oxoGuo).**

| Follow-up combination | Number of patients | preoperative sample | Postoperative samples | | | | Samples per patient | Number of patients |
|---|---|---|---|---|---|---|---|---|
| | | | 3 months | 6 months | 12 months | 24 months | | |
| 1 | 69 | 1 | 1 | 1 | 1 | 1 | 5 | 69 |
| 2 | 16 | 1 | 1 | 1 | 0 | 1 | 4 | 117 |
| 3 | 21 | 1 | 1 | 0 | 1 | 1 | | |
| 4 | 20 | 1 | 0 | 1 | 1 | 1 | | |
| 5 | 60 | 1 | 1 | 1 | 1 | 0 | | |
| 6 | 8 | 1 | 1 | 0 | 0 | 1 | 3 | 101 |
| 7 | 14 | 1 | 0 | 0 | 1 | 1 | | |
| 8 | 9 | 1 | 0 | 1 | 0 | 1 | | |
| 9 | 22 | 1 | 1 | 0 | 1 | 0 | | |
| 10 | 10 | 1 | 0 | 1 | 1 | 0 | | |
| 11 | 38 | 1 | 1 | 1 | 0 | 0 | | |
| 12 | 11 | 1 | 0 | 0 | 0 | 1 | 2 | 69 |
| 13 | 13 | 1 | 0 | 0 | 1 | 0 | | |
| 14 | 10 | 1 | 0 | 1 | 0 | 0 | | |
| 15 | 35 | 1 | 1 | 0 | 0 | 0 | | |
| **Total** | **356** | **356** | **269** | **232** | **229** | **168** | | **356** |

100 and sixty-nine, it could be calculated at three, two and one timepoint postoperative, respectively (Table 1). Twelve months or later after RYGB, follow up excretion estimates could be calculated for 273 (76%) of the included patients. For 8-oxodG, numbers were similar. On average, the preoperative samples were collected 4.56 (4.19–4.93) months (mean with a 95% CI) before surgery. The postoperative samples were collected 3.12 (3.07–3.17), 6.38 (6.27–6.49), 12.31 (12.16–12.46) and 24.58 (24.28–24.88) months after RYGB surgery, respectively.

All urine and blood samples were frozen shortly after sampling at −80˚C and stored between three and eight years at the time for analysis of 8-oxodG and 8-oxoGuo.

Clinical data like for example surgery date, bodyweight, blood pressure, as well as smoking status and information about anti-diabetic, anti-hypertensive and lipid-lowering treatment, had been collected in a database at pre- and postoperative consultations and were later validated by cross-check in medical journals and records of prescribed medicine. Information about gender was extracted from the Danish social security number given to every resident in Denmark. Biochemical data from 2009 and onwards, including results from routine blood- and urine analyses, were retrieved from the laboratory information system. In this paper, we have included biochemical data until May 2017. Depending on time for enrolment and on variation in adhesion to postoperative follow-up, some of the patient data records, either clinical and/or biochemical were not complete for all patients.

This study was performed in accordance with the Helsinki Declaration and was approved by the scientific Ethics Committee of the Capital Region, Denmark, protocols number HD2009-78 and H-6-2014-029, and by the Danish Data Protection Agency. All participants in the study gave written informed consent.

## Diabetes subgroups

Biochemical data, together with data on antidiabetic medication, were used to divide patients in subgroups according to diabetes status, as illustrated in Fig 1. Diabetes was defined as presence in biochemical records of hyperglycemia (HbA$_{1c}$ > 48 mmol/mol (6.5%), plasma

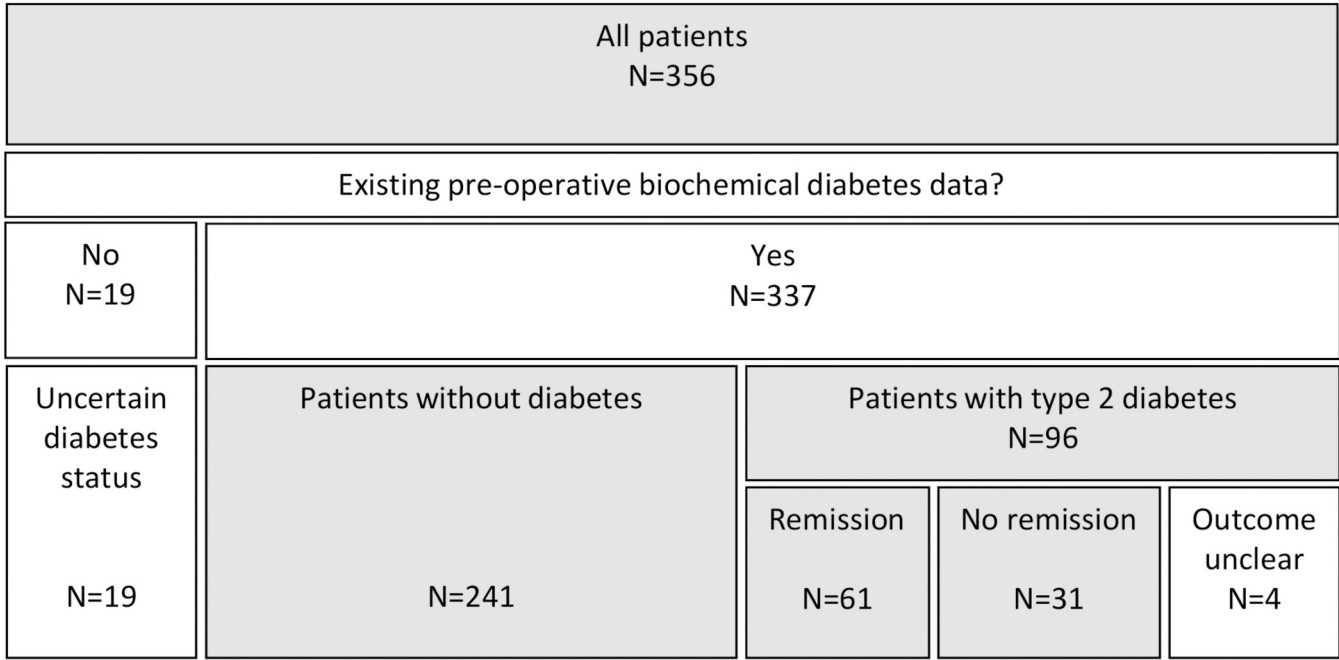

**Fig 1. Subdivision according to diabetes status.** Marked with grey color are the groups discussed in this paper. NDH, patients without diabetes; NDH, patients with type 2 diabetes; DH-NDH, patients with diabetes who obtained remission in diabetes after RYGB; DH-DH, patients with diabetes who did not obtain remission in diabetes after RYGB. Twenty-three patients did not fit in any of the groups NDH, DH-NDH or DH-DH.

glucose > 7.0 mmol in the fasting state or > 11.1 mmol as 120 minutes-value in a 75 g oral glucose tolerance test). Remission of diabetes after RYGB was recognized as described previously [35] in patients with diabetes who after RYGB were off antidiabetic medicine, with $HbA_{1c}$ staying below 48 mmol/mol for the entire remaining span of their biochemical records. In tables and figures, the subgroup of patients with diabetes is called DH (short for diabetes and hyperglycemia). NDH is the group without diabetes. The group of patients who obtained remission is called DH-NDH. The patients in group DH-DH did not obtain remission of diabetes.

Of the 356 patients, 337 had biochemical data both before and after surgery, ninety-six (27%) of these had type 2 diabetes. Sixty-one (64%) of the patients with type 2 diabetes passed the criteria for remission of diabetes after RYGB, described above. Twenty-three of the patients, four of which had type 2 diabetes, did not fit in any of the three groups NDH, DH-NDH, DH-DH (Fig 1).

### Analysis of 8-oxodG and 8-oxoGuo

The analytical method for measuring concentrations of the urinary markers of nucleic acid damage—liquid chromatography coupled with tandem mass spectrometry—has been described in detail [36]. Half of the samples were analyzed in 2015–2016 and the rest in 2018. To be sure that levels were comparable, analyses of 310 random samples analyzed in 2015–2016 were repeated in 2018. Mean coefficient of variation (CV) was 14% and 13%, for 8-oxodG and 8-oxoGuo, respectively, when comparing 2018- to 2015–2016 levels. In samples analyzed 2018, urinary concentrations of 8-oxodG and 8-oxoGuo were approximately 4–5 nmol/L and 6 mmol/L higher, respectively. Samples analyzed in 2015–2016, compared to samples analyzed in 2018, had a similar distribution of preoperative and postoperative samples. Both in 2015–2016 and in 2018, analyses have been stable with a day to day CV of less than 8%

and a mean CV at 4–5% for double estimations. Only three results for 8-oxodG (none for 8-oxoGuo) could not be reported due to interference in the chromatograms.

Traditionally, to reduce variability of analyte concentration in spot urine samples from variation in urine volume, concentrations of 8-oxodG and 8-oxoGuo were normalized to urinary creatinine concentrations. We analyzed urine creatinine with an in-house assay (Jaffé-method [37]) that has a day to day CV of 3.1%, a CV at 2.8% for double estimations and a trueness at 114% corresponding to the commercial assays (assessed through external quality control). The in-house procedure is half-automatized, using a Biomek robot from Beckman Coulter for sample preparation on a 96-well microplate. The absorbance is measured with a Multiskan FC Microplate Photometer from Thermo Scientific. Creatinine concentrations measured in 218 preoperative urine samples were seen to correlate well to the results from a Cobas 6000 from Roche, measured on the day of sampling, with $R^2 = 0.96$ and a mean CV at 4.9%.

Urinary creatinine alone did not, however, suffice as normalization factor in this study, the reasons are explained in the section below.

**Using a physiological model to adjust for changes in muscle mass.** Apart from a large loss of fat mass, weight loss after RYGB includes a substantial decrease (12–16%) in muscle mass [38]. A similar decrease is seen in the excretion rate of urinary creatinine [39]. Decreases in 24-hour urinary creatinine with 18–24% at six and twelve months postoperative follow up after RYGB, compared to preoperative mean values of 1.3 g, have been reported [40, 41]. In a subsample of nineteen patients from our own RYGB-cohort, lean limb mass (a surrogate measure of skeletal muscle mass) was reduced with 11.4% six months after RYGB [42]. The changes in muscle mass after RYGB and the assumed variation they impose on urinary creatinine in our study population challenge the traditional normalization-approach with urinary creatinine mentioned in the previous section, as excretion of creatinine is not constant over time in these patients.

To face this challenge, we used a physiological model to calculate estimates of 24-hour excretion of the urinary markers—an approach that has been described in detail and validated in a recent study [43]–for all pre- and postoperative samples. To calculate estimates of GFR, we used the CKD-EPI Cystatin C equation from 2012 in combination with the common formula by Du Bois & Du Bois for calculation of body surface area (BSA). Serum cystatin C was measured on a Cobas 6000 c 501 module (Tina-quant Cystatin C immunoturbidimetric method), Roche Diagnostics. The cystatin C based GFR-estimate was unaffected by weight loss (and hence muscle mass) in the above mentioned study on nineteen RYGB patients from our own cohort [42].

## Other biochemical variables

All other blood-, plasma- and urine parameters were routine biochemical analyses, measured as described previously [33]. Routine eGFR was calculated on the day of sampling with the CKD-EPI Creatinine equation from 2009. Results reported as > 90 mL/min in the laboratory data system were counted as 90 mL/min.

Estimations of insulin resistance according to the Homeostasis Model Assessment (HOMA-IR) were performed in the HOMA2-calculator software, version 2.2.3, using concentrations of Plasma-glucose and C-peptide from the same time of sampling.

Due to a systematical bias in Scandinavian laboratories (including ours) measuring $HbA_{1c}$ with high pressure liquid chromatographic methods, $HbA_{1c}$-results from before January 2013 were adjusted with– 2.7 mmol/mol (0.25%) as previously described [35] to allow for comparisons of levels on both sides of the calibrator change. The NGSP converter converted HbA1c-values in mmol/mol to %.

## Data analysis and statistics

Based on previous controlled trials with power calculations [44] we evaluated that the study could be conducted with high power and thus acceptable statistical certainty. All differences, both within the same group (comparisons of pre- and postoperative levels) and between groups have been calculated *post hoc* to have a power above 0.9 when effect-sizes exceeds 0.8.

Estimates of pre- and postoperative 24-hour 8-oxoGuo and 8-oxodG excretion were calculated as described above.

Normal distribution, evaluated by inspection of a q-q plot, was seen for most variables, which with few exceptions allowed for the use of parametric statistics.

The two urinary markers showed a slight tendency, similar in subgroups, towards light left skewed distributions. The skew was not large enough, however, to abandon parametric statistics. A repeated measures ANOVA combined with Mauchly's test of sphericity was used, with group mean substitution for missing values (528 out of 1780 datapoints for 8-oxodG and 526 for 8-oxoGuo were imputed); followed by a Greenhouse-Geisser correction for violation of sphericity. Post-operative results were compared with the pre-operative with planned contrast analyses. To be sure not to over-interpret post-operative changes, a Bonferroni correction was applied. Extreme values (identified by inspection of a boxplot) were few and removing them did not alter significance, just as correction for the reduced degrees of freedom due to imputation did not.

Because of the unequal gender distribution in our subgroups and because we know, since long, that gender has an influence on the urinary marker levels [45], data from female and male patients were analyzed separately. In the larger female population, differences between patients with and without diabetes and between diabetes subgroups were detected using unpaired t-tests and one-way ANOVA with contrast analysis, respectively.

Smoking and lipid-lowering treatment are other possible confounders [46]. To exclude tentative changes in smoking status or in lipid-lowering drug intake during the time of follow up as a reason for a change in urinary marker levels, we also choose to evaluate possible differences between smokers and non-smokers as well as patients on and not on lipid-lowering drugs, by visual interpretation of graph.

Correlations between the urinary markers and other variables (including the urinary markers normalized in a traditional manner to urinary creatinine) were investigated with Spearman's correlation.

Statistic calculations were performed in the IBM SPSS Statistics software, versions 22 and 25.

## Results

### Preoperative characteristics

The mean age in the study population was 44.2 years, BMI was 42.1 kg/m$^2$ and 69.1% of the patients were females. Compared to patients without diabetes, patients with type 2 diabetes were older, with a slightly lower BMI and to a larger part male. A selection of clinical variables is presented below, in Table 2 for patients without diabetes and patients belonging to either of the two main diabetes subgroups discussed in this paper (N = 337). Detailed preoperative clinical and biochemical characteristics are reported in S1 Table for the entire research population (N = 356). Here, it is shown that patients with diabetes had lower cholesterol-levels (consistent with the high number taking cholesterol-lowering drugs) than the patients without diabetes. BSA, cystatin C and estimates of kidney function (eGFR Creatinine and eGFR Cystatin C) were comparable between patients with and without diabetes and between diabetes subgroups,

**Table 2. Preoperative characteristics for the three main study population subgroups.**

| | NDH Patients without diabetes | DH-NDH Patients with type 2 diabetes, in remission after RYGB | DH-DH Patients with type 2 diabetes with persisting hyperglycemia after RYGB | ANOVA P-value |
|---|---|---|---|---|
| Number of patients in the group (N) | 241 | 61 | 31 | |
| Age (years) | 41.7 (9.1) | 49.1 (9.1) | 50.8 (6.9) | < 0.0001 |
| Females (%) | 75.5 | 54.0 | 55.0 | 0.003 |
| Bodyweight (kg) | 124.9 (22.5) | 125.9 (22.6) | 115.8 (17.6) | 0.078 |
| BMI (kg/m$^2$) | 42.7 (5.6) | 41.8 (5.9) | 40.0 (3.9) | 0.029 |
| Present smoker (%) | 17.5 | 22.9 | 6.5 | 0.044 |
| Lipid-lowering treatment (%) | 10.0 | 51.0 | 77.0 | < 0.0001 |

Data are reported as Mean (SD), except for the parameters reported as a percentage or number. Age is on the day of Roux-en-Y gastric bypass (RYGB) surgery. Data represent the closest available data point before surgery. NDH, patients with biochemical glucose markers below diagnostic threshold for diabetes and not on antidiabetic treatment; DH-NDH, patients who obtained remission of diabetes after RYGB; DH-DH, patients who did not obtain remission in diabetes after RYGB.

and all patient groups had a mean blood pressure within the normal range (consistent with an overall high prevalence of anti-hypertensive treatment).

## Preoperative excretion of 8-oxodG and 8-oxoGuo

Mean 24-hour excretion levels of 8-oxodG and 8-oxoGuo, in urine sampled before surgery are reported in detail in Table 3, for both genders and patients with and without type 2 diabetes. In brief, excretion of 8-oxoGuo was higher than excretion of 8-oxodG. There was a highly significant difference between female and male patients, male patients having 40–50% higher excretion levels. In the female subpopulation, 24-hour excretion of 8-oxodG was marginally lower in patients with diabetes than in patients without diabetes, while no difference was seen between patients with and without diabetes for 8-oxoGuo.

**Table 3. Preoperative excretion of 8-oxodG and 8-oxoGuo for female and male patients with and without diabetes.**

| | | preoperative 24-hour 8-oxodG (nmol) | | | | | DH vs. NDH |
|---|---|---|---|---|---|---|---|
| | all | | NDH | | DH | | |
| | n | mean (95% CI) | n | mean (95% CI) | n | mean (95% CI) | P |
| All | 356 | 18.9 (18.0–19.7) | 241 | 19.1 (18.1–20.1) | 96 | 18.5 (16.7–20.3) | 0.536 |
| Females | 246 | 16.8 (16.0–17.6) | 182 | 17.3 (16.3–18.2) | 52 | 15.2 (13.3–17.1) | 0.044 |
| Males | 110 | 23.4 (21.7–25.2) | 59 | 24.7 (22.3–27.1) | 44 | 31.4 (28.7–34.2) | 0.208 |
| F vs. M | P | < 0.0001 | | < 0.0001 | | < 0.0001 | |
| | | preoperative 24-hour 8-oxoGuo (nmol) | | | | | DH vs. NDH |
| | all | | NDH | | DH | | |
| | n | mean (95% CI) | n | mean (95% CI) | n | mean (95% CI) | P |
| All | 356 | 24.4 (23.5–25.3) | 241 | 23.8 (22.8–24.9) | 96 | 26.0 (24.1–27.9) | 0.036 |
| Females | 246 | 21.0 (20.2–21.7) | 182 | 20.8 (20.0–21.6) | 52 | 21.4 (19.5–23.3) | 0.534 |
| Males | 110 | 32.1 (30.5–33.7) | 59 | 33.1 (31.0–35.2) | 44 | 31.4 (28.7–34.2) | 0.324 |
| F vs. M | P | < 0.0001 | | < 0.0001 | | < 0.0001 | |

Data are reported as mean with a 95% confidence interval. NDH, patients with biochemical glucose markers below diagnostic threshold for diabetes and not on antidiabetic treatment; DH, patients with biochemically confirmed diabetes. All patients also include patients for whom we were not able to confirm diabetes status. Exact P-values from unpaired t-tests are reported down to 0.001.

A table of preoperative excretion levels of 8-oxodG and 8-oxoGuo, normalized to urinary creatinine in the traditional manner, can be found in S2 Table.

## 24-hour excretion of 8-oxodG and 8-oxoGuo after RYGB

After RYGB, a repeated measures ANOVA with group mean substitution of missing post-operative values showed statistically significant changes in the two urinary markers, $F$ (3.748, 1330.535) = 68.014, $P < 0.0001$ and $F$ (3.601, 1278.454) = 52.668, $P < 0.0001$ for 8-oxodG and 8-oxoGuo, respectively. Also when missing post-operative values were replaced with the individual pre-operative value, instead of the group mean, results were highly significant ($F$ (3.669, 1302.350) = 20.71, $P < 0.0001$ for 8-oxodG and $F$ (3.742, 1328.309) = 10.82, $P < 0.0001$ for 8-oxoGuo). As shown in Fig 2A, a temporary increase was seen for 8-oxodG three months after RYGB, compared to excretion levels before surgery. After the initial increase, levels of 8-oxodG decreased during the rest of the follow-up period, and two years after RYGB, excretion levels were approximately reduced by 12% compared to preoperative levels (Fig 2A). For 8-oxoGuo, we found a similar, although more gradual decrease from three months and onwards without the initially raised level that was seen for 8-oxodG (Fig 2B). The same patterns were seen for patients with and without diabetes (Fig 3A & 3B). It was also seen for female as well as male patients, for smokers as well as non-smokers, and for patients on and not on cholesterol-lowering drugs.

In Fig 3A, data from the female subpopulation show that the trend towards lower excretion levels of 8-oxodG in patients with diabetes was consistent throughout the postoperative follow-up period. A similar trend was seen in the smaller male subpopulation. For 8-oxoGuo, postoperative 24-hour excretion levels did not differ between patients with and without diabetes (Fig 3B). The largest difference in excretion levels for 8-oxodG between patients with and without diabetes was seen between patients without diabetes and the group of patients with type 2 diabetes who did not obtain remission in diabetes after RYGB (Fig 3C). This group of patients with diabetes had between 6–7 nmol lower 24-hour excretion level of 8-oxodG, than patients without diabetes. Also, there seemed to be an approximate 4–6 nmol difference between this group and the other group of patients with diabetes (Fig 3C). Between the group of patients who obtained remission in their type 2 diabetes after RYGB and patients without diabetes, no difference was seen.

## Bodyweight, BMI and markers of glucose and lipid metabolism after RYGB

Relative changes in body weight, BMI and markers of glucose metabolism are roughly outlined in Fig 4. As expected after RYGB (and previously described in this study population [33]), there was a rapid loss of bodyweight, with mean BMI dropping approximately ten units during the first six months, in both patients with and without type 2 diabetes. Thereafter, mean weight was stabilized between 88–95 kg. The markers of glucose metabolism (HbA$_{1c}$, fasting plasma concentrations of glucose, insulin and C-peptide) were improved to a large extent already at three months after the surgery—and HOMA-IR followed. Three months after RYGB, mean HbA$_{1c}$ levels in this sample of the study population had decreased from 39.2 (38.0–40.3) mmol/mol to 34.9 (34.1–35.8) mmol/mol, glucose had decreased from 6.2 (6.0–6.5) mmol/L to 5.7 (5.5–5.8) mmol/L, insulin concentrations were halved from 123 (113–133) pmol/L to 63 (59–67) pmol/L and C-peptide fell from 1234 (1181–1287) pmol/L to 905 (865–946) pmol/L. Mean HOMA-IR fell from 3.0 (2.8–3.1) to a value of 2.1 (2.0–2.2) three months after RYGB. Improvements were also seen for the traditionally measured plasma lipids. Twelve months postoperative there had been a 30% mean increase in HDL-cholesterol and a 25% and 40% decrease in LDL-cholesterol and triglycerides, respectively.

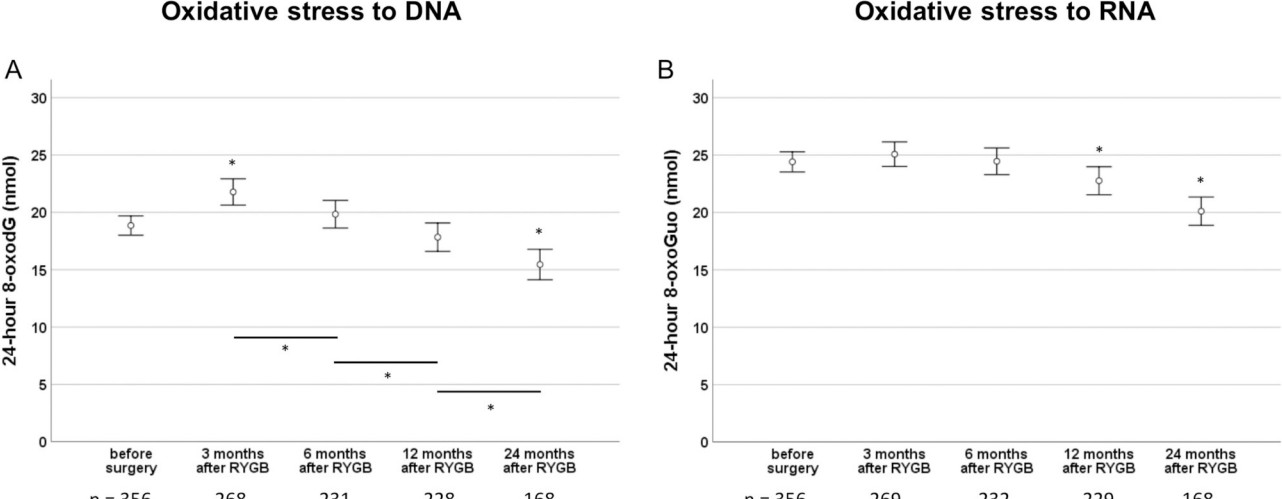

**Fig 2. Changes in oxidative stress to DNA and RNA after Roux-en-Y gastric bypass surgery.** The figures show 24-hour pre- and postoperative excretion of 8-oxodG (Fig A) and 8-oxoGuo (Fig B) in the whole study population. Data are reported as means with error bars showing the 95% confidence interval of the mean and * symbolizes a significant difference after Bonferroni correction. The number of patients, n, at each time point is shown at the bottom of each figure.

## Correlations between 8-oxodG, 8-oxoGuo and other variables

Correlation coefficients before and after RYGB, between 8-oxodG, 8-oxoGuo, BMI and markers of glucose- and lipid metabolism, are shown in Table 4.

For both 8-oxodG and 8-oxoGuo, there was a strong positive correlation between the traditionally urinary creatinine normalized and the calculated 24-hour excretion levels before as well as after surgery.

BMI correlated positively to 8-oxoGuo both before and after surgery, significantly in both patients with and without diabetes, while no correlation was seen between the absolute values of BMI and 8-oxodG. Between delta values, representing the difference between pre- and postoperative BMI and 8-oxodG, respectively, there were, however, a weak but significant negative correlation after RYGB. The largest reductions in both urinary markers were seen in patients with a moderate weight loss, as illustrated in S2 Fig, where individual patient changes in 8-oxodG and 8-oxoGuo 24 months after RYGB are shown in relation to relative postoperative BMI.

Although no correlation was found in the entire study population between $HbA_{1c}$ and 8-oxoGuo, there was a weak negative correlation between $HbA_{1c}$ and 8-oxoGuo twelve and twenty-four months after RYGB in patients without diabetes. $HbA_{1c}$ correlated negatively to 8-oxodG both before and after surgery, strongest after RYGB in the group of patients without diabetes. Plasma insulin concentration correlated positively to 8-oxoGuo both before and after RYGB and did not correlate to 8-oxodG. Also, C-peptide and HOMA-IR (but not glucose), showed weak positive correlations to 8-oxoGuo before surgery, as well as twenty-four months after RYGB. C-peptide, HOMA-IR and glucose showed no correlations to 8-oxodG. HDL-cholesterol correlated negatively pre- and postoperative to both 8-oxoGuo and 8-oxodG, while no correlation was seen between any of the urinary markers and LDL-cholesterol. Twelve months after RYGB, delta values of HDL-cholesterol and delta values of triglycerides showed negative and positive correlations, respectively, to 8-oxoGuo.

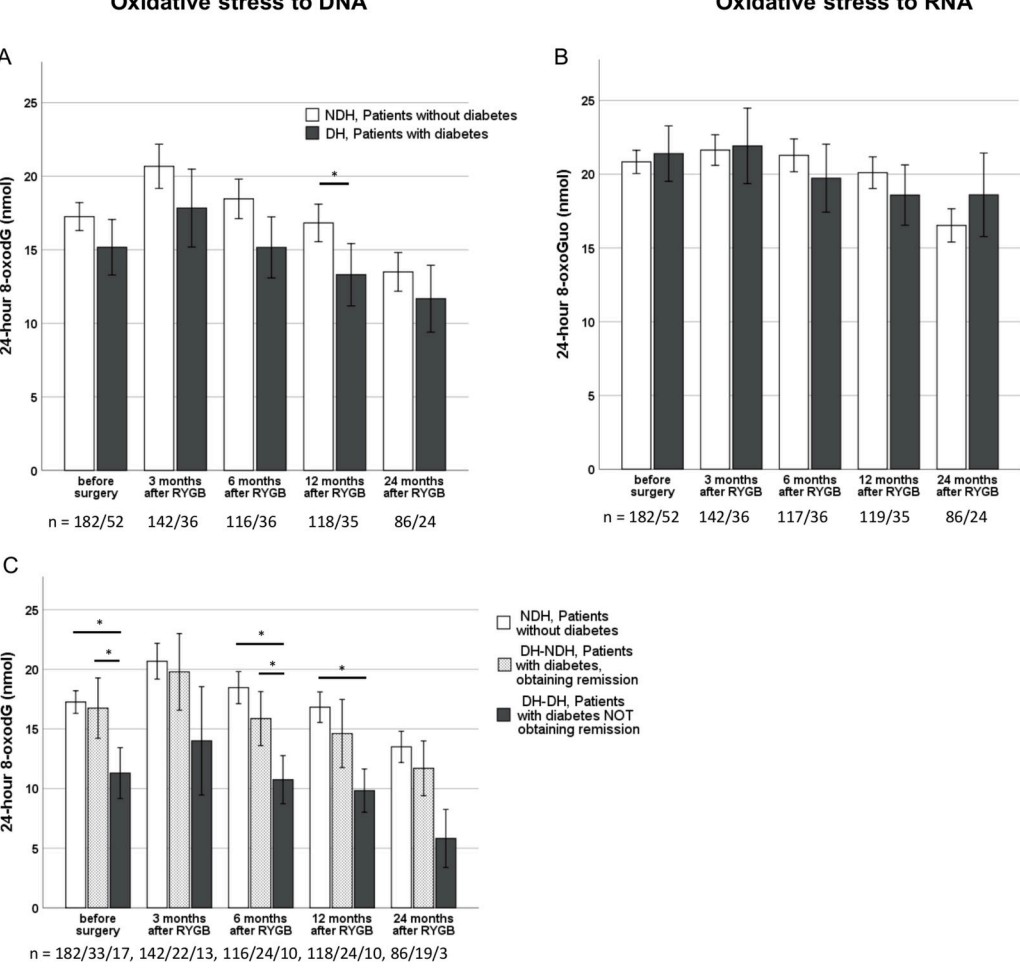

**Fig 3. Differences in oxidative stress to DNA and RNA depending on diabetes status.** The Figs A & B report data for the female subpopulation divided in two subgroups: patients without diabetes (white bars) and patients with diabetes (black bars). Fig C reports data for the female subpopulation divided in three subgroups: NDH, patients without diabetes (white bars); DH-NDH, patients with diabetes who obtained remission in diabetes after RYGB (dotted bars) and DH-DH, patients with diabetes who did not obtain remission in diabetes after RYGB (black bars). Data are reported as means with error bars showing the 95% confidence interval of the mean and * symbolizes a significant difference after Bonferroni correction. The number of patients, n, at each time point is shown at the bottom of each figure.

## Discussion

To our knowledge, this study is the first to describe changes in excretion rates of 8-oxodG and 8-oxoGuo, two urinary markers of DNA and RNA oxidation, after RYGB surgery in humans. A main finding was an increase in oxidative stress on DNA the first months after surgery, evidenced by increased excretion of 8-oxodG. In the oxidative stress on RNA, for which 8-oxoGuo excretion serves as a quantitative measure; there was a gradual reduction after surgery and significantly reduced levels twelve months postoperative. Two years after RYGB and a mean weight loss of thirty-five kg and a markedly improved glucose metabolism, including a reduced insulin resistance, excretion rates of both 8-oxodG and 8-oxoGuo had decreased well below preoperative levels.

From this, we conclude, that a decrease in oxidative stress on nucleic acids might be contributing to the many long-term benefits of RYGB-surgery in obesity and type 2 diabetes but is

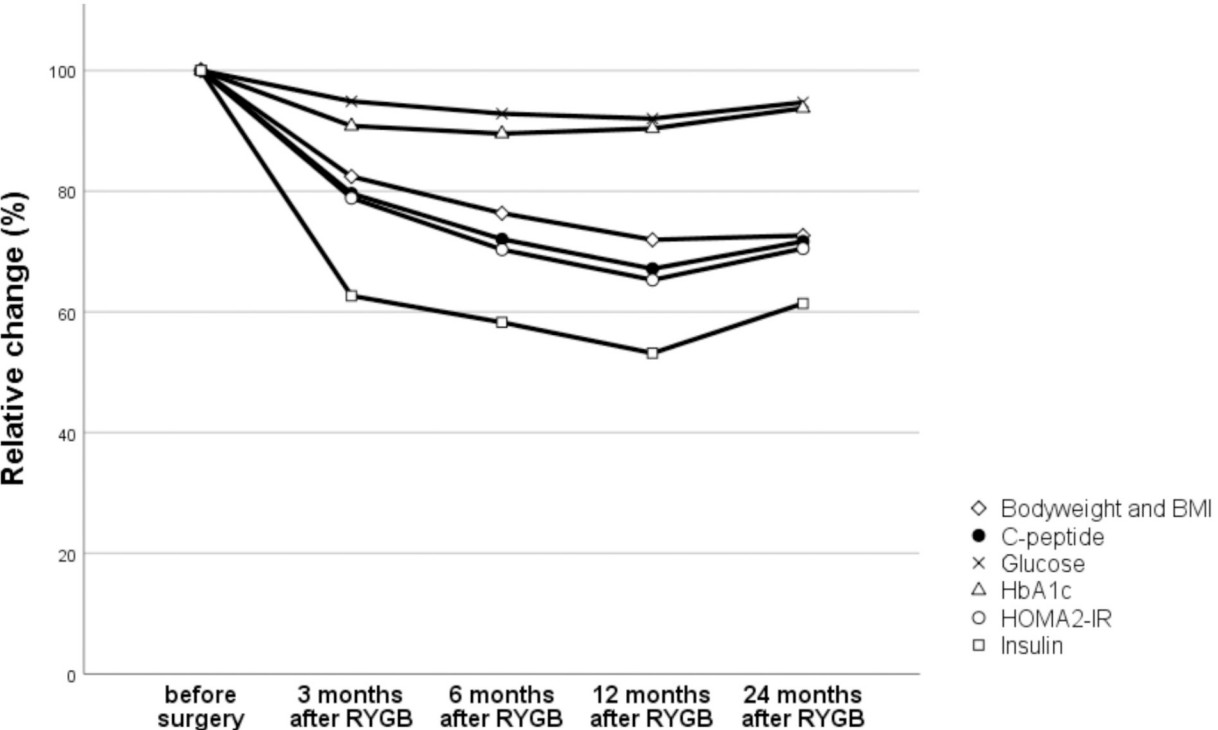

**Fig 4. Relative changes in body weight, BMI and markers of glucose metabolism after RYGB.** Data are means of levels relative to the preoperative means (in percent) for body weight and BMI (diamonds), C-peptide (black circles), fasting glucose (x'es), HbA1c (triangles), HOMA2-IR (white circles) and Insulin (squares).

less likely a main contributing factor to the immediate effects of RYGB-surgery on weight, glucose- and lipid metabolism within the first postoperative weeks, primarily explained by an increase in insulin sensitivity of the liver [22, 47]. After months and major weight loss also peripheral insulin sensitivity is improved [47]. This is interesting, as a reduced oxidative stress to DNA has been suggested to be one of the acute mechanisms of action of sleeve gastrectomy [14].

We observed a transient increase in 8-oxodG after surgery. Surgery itself is a stressful event with many physiological responses. We cannot decipher specific causes for the increased oxidative stress measures after surgery, but expect that several factors are in play, and that they take some time to normalize after surgery and physiological resetting and reduction of food intake.

The reasons why metabolic diseases or the obese condition lead to increased formation of 8oxodG and 8oxoGuo are not known. In isolated obesity without complications, we previously reported that only 8oxoGuo is increased [15], indicating that obesity by itself increases oxidative stress by an unknown mechanism. We suggest that the obese condition is a metabolic stress to the mitochondria and that several physiological responses to obesity and overeating, for instance the release of adipokines, together make the mitochondrial respiratory chain slightly less efficient—not in the context of energy production, but in the context of increased production of ROS such as the superoxide anion, hydrogen peroxide and the hydroxy radical, that in turn will oxidize the RNAs in the vicinity of the mitochondria. RNA will in this situation function as a "photographic plate" for mitochondrial ROS production. Also, oxidized

**Table 4. Correlations between 8-oxodG, 8-oxoGuo, BMI and markers of glucose and lipid metabolism.**

| | | 24-hour 8-oxodG (nmol) | | | | 24-hour 8-oxoGuo (nmol) | | | |
|---|---|---|---|---|---|---|---|---|---|
| | | before surgery | 12 months after RYGB | 24 months after RYGB | Δ:Δ 12 months after RYGB | before surgery | 12 months after RYGB | 24 months after RYGB | Δ:Δ 12 months after RYGB |
| 24-hour 8-oxoGuo | $r_s$ | 0.643** | 0.692** | 0.645** | | | | | |
| | P | < 0.0001 | < 0.0001 | < 0.0001 | | | | | |
| | n | 356 | 228 | 168 | | | | | |
| 8-oxodG (nmol/mmol creatinine) | $r_s$ | 0.755** | 0.849** | 0.837** | | | | | |
| | P | < 0.0001 | < 0.0001 | < 0.0001 | | | | | |
| | n | 356 | 228 | 168 | | | | | |
| 8-oxoGuo (nmol/mmol creatinine) | $r_s$ | | | | | 0.578** | 0.745** | 0.769** | |
| | P | | | | | < 0.0001 | < 0.0001 | < 0.0001 | |
| | n | | | | | 356 | 229 | 168 | |
| BMI, All | $r_s$ | 0.032 | 0.075 | 0.047 | -0.189** | 0.137** | 0.207** | 0.235** | -0.057 |
| | P | 0.553 | 0.262 | 0.546 | 0.004 | 0.009 | 0.002 | 0.002 | 0.391 |
| | n | 356 | 228 | 168 | 228 | 356 | 229 | 168 | 229 |
| BMI, NDH | $r_s$ | 0.063 | 0.128 | 0.087 | -0.159 | 0.189** | 0.244** | 0.144 | -0.071 |
| | P | 0.330 | 0.118 | 0.366 | 0.051 | 0.003 | 0.002 | 0.131 | 0.389 |
| | n | 241 | 150 | 111 | 150 | 241 | 151 | 111 | 151 |
| BMI, DH | $r_s$ | 0.033 | 0.068 | 0.092 | -0.220 | 0.217* | 0.234 | 0.426** | 0.004 |
| | P | 0.753 | 0.590 | 0.538 | 0.078 | 0.034 | 0.061 | 0.003 | 0.972 |
| | n | 96 | 65 | 47 | 65 | 96 | 65 | 47 | 65 |
| HbA1c, All | $r_s$ | -0,112* | -0.282** | -0.319** | -0.68 | 0.086 | -0.093 | -0.003 | -0.035 |
| | P | < 0.037 | < 0.0001 | < 0.0001 | 0.317 | 0.109 | 0.166 | 0.965 | 0.601 |
| | n | 348 | 225 | 164 | 219 | 348 | 226 | 164 | 220 |
| HbA1c, NDH | $r_s$ | -0.103 | -0.270** | -0.414** | -0.119 | 0.031 | -0.188* | -0.241* | -0.061 |
| | P | 0.115 | 0.001 | < 0.0001 | 0.155 | 0.637 | 0.021 | 0.012 | 0.463 |
| | n | 235 | 148 | 109 | 144 | 235 | 149 | 109 | 145 |
| HbA1c, DH | $r_s$ | -0.031 | -0.245 | -0.297* | -0.166 | -0.061 | -0.069 | 0.063 | -0.112 |
| | P | 0.770 | 0.051 | 0.048 | 0.196 | 0.562 | 0.590 | 0.681 | 0.385 |
| | n | 94 | 64 | 45 | 62 | 94 | 64 | 45 | 62 |
| Insulin | $r_s$ | 0.044 | 0.051 | 0.034 | -0.104 | 0.193** | 0.155* | 0.208* | -0.075 |
| | P | 0.416 | 0.457 | 0.677 | 0.139 | < 0.001 | 0.023 | 0.01 | 0.283 |
| | n | 346 | 213 | 152 | 205 | 346 | 214 | 152 | 206 |
| Glucose | $r_s$ | -0.088 | -0.128 | -0.131 | -0.116 | 0.062 | 0.055 | 0.072 | -0.070 |
| | P | 0.098 | 0.057 | 0.095 | 0.813 | 0.248 | 0.416 | 0.361 | 0.302 |
| | n | 351 | 223 | 164 | 220 | 351 | 224 | 164 | 221 |
| C-peptide | $r_s$ | 0.020 | -0.040 | 0.017 | -0.064 | 0.170** | 0.089 | 0.186* | -0.011 |
| | P | 0.709 | 0.549 | 0.831 | 0.345 | 0.001 | 0.183 | 0.017 | 0.875 |
| | n | 349 | 225 | 164 | 220 | 349 | 226 | 164 | 221 |
| HOMA-IR | $r_s$ | -0.008 | -0.059 | -0.002 | -0.043 | 0.158** | 0.101 | 0.203** | -0.012 |
| | P | 0.879 | 0.380 | 0.977 | 0.542 | 0.004 | 0.134 | 0.009 | 0.868 |
| | n | 336 | 221 | 162 | 207 | 336 | 222 | 162 | 208 |
| Total Cholesterol | $r_s$ | 0.034 | -0.037 | | -0.008 | 0.010 | -0.133* | | 0.094 |
| | P | 0.527 | 0.582 | | 0.907 | 0.846 | 0.045 | | 0.158 |
| | n | 354 | 227 | | 226 | 354 | 228 | | 227 |
| HDL-Cholesterol | $r_s$ | -0.150** | -0.188** | | -0.111 | -.193** | -0.145* | | -0.181** |
| | P | 0.005 | 0.004 | | 0.097 | < 0.001 | 0.029 | | 0.006 |
| | n | 354 | 227 | | 226 | 354 | 228 | | 227 |

(*Continued*)

**Table 4.** (Continued)

| | | 24-hour 8-oxodG (nmol) | | | | 24-hour 8-oxoGuo (nmol) | | | |
|---|---|---|---|---|---|---|---|---|---|
| | | before surgery | 12 months after RYGB | 24 months after RYGB | Δ:Δ 12 months after RYGB | before surgery | 12 months after RYGB | 24 months after RYGB | Δ:Δ 12 months after RYGB |
| LDL-Cholesterol | $r_s$ | 0.103 | 0.055 | | 0.017 | 0.023 | -0.072 | | 0.112 |
| | $P$ | 0.054 | 0.405 | | 0.798 | 0.664 | 0.281 | | 0.097 |
| | n | 349 | 227 | | 221 | 349 | 228 | | 222 |
| Triglycerides | $r_s$ | 0.002 | 0.061 | | 0.040 | 0.133* | 0.060 | | 0.205** |
| | $P$ | 0.971 | 0.357 | | 0.547 | 0.012 | 0.366 | | 0.002 |
| | n | 354 | 227 | | 226 | 354 | 228 | | 227 |

Exact $P$-values are reported down to 0.001. Below 0.001, $P$-values are reported as <0.001 or <0.0001.

* and ** indicates significance on the 0.05- and 0.01-level, respectively.

For BMI and HbA1c, correlations are shown for the whole study population and for the subpopulations of patients with (DH) and without diabetes (NDH). For BMI, HOMA-IR and lipids, correlations between the delta values 12 months after Roux-en-Y gastric bypass (RYGB) are shown in the columns marked Δ:Δ for 8-oxodG and 8-oxoGuo, respectively.

RNAs have been shown to have aberrant functions [7] with implications for disease processes [5].

A reduction in bodyweight has been reported to reduce other markers of oxidative stress [48, 49], but so far, only few interventions except for smoking cessation, olive oil intake [46] and perhaps sleeve gastrectomy [14] have been able to reduce levels of urinary markers of DNA or RNA oxidation. Nevertheless, since ROS clearly play a role in pathogenesis of various diseases, (focus has been on the toxicological aspects of high levels of oxidative stress, however it is now realized that modifications in the redox balance is a multifaceted regulatory mechanism in normal physiological situations [5]), preventive and therapeutic strategies that aim to restore redox homeostasis may be an intuitive approach with a large potential to protect against metabolic diseases [3–5]. The present study showed that RYGB-induced weight loss does indeed reduce oxidative stress on DNA and RNA. The changing physiology following RYGB might therefore provide a way to understand the mechanisms of action that improve redox-homeostasis.

In patients with persisting hyperglycemia after RYGB, excretion rates of 8-oxodG were lower after surgery than for patients without type 2 diabetes, independently of gender and smoking status. It is not clear, whether the reason for this difference is a smaller increase in oxidant formation after RYGB in the group of patients who did not respond to the surgery with diabetes remission and therefore continued diabetes treatment; if the proportion of anti-oxidant defense mechanisms, for example dietary or enzymatic anti-oxidants, might be higher in this group; or if DNA's availability for oxidation by some reason might be lower. A higher $HbA_{1c}$ in the group of patients without diabetes was also associated to lower levels of 8-oxodG after RYGB, suggesting that changes in 8-oxodG after RYGB somehow are modified by glucose metabolism.

An association between 8-oxoGuo and diabetes has been described in more than one population [16, 50]. These studies studied excretion levels of 8-oxoGuo, normalized to urinary creatinine. In our obese study population, we could not confirm the presence of a similar association after adjusting for differences in weight, height and kidney function—the variables that were included in the physiological model for estimating 24-hour excretion. Although there was no convincing association between 24-hour 8-oxoGuo excretion and $HbA_{1c}$, we did find higher 8-oxoGuo excretion levels with increasing insulin, c-peptide and HOMA-IR,

indicating a connection between 8-oxoGuo and the hyper-insulinemic, insulin resistant state of metabolic disease, which in this morbidly obese study population was not exclusive for patients with diabetes. These findings also suggest that bodyweight, height and perhaps kidney function should be taken into consideration and corrected for, when urinary levels of oxidative damage to nucleic acids are compared between study populations. This appears rational, as the size and composition of the body have impact on the metabolic rate that affect the production of ROS in the mitochondria [51] and that there is a direct association between ROS-formation and oxidative damage to DNA and RNA [5]. Although the exact source of ROS that oxidize DNA is not known in detail, it is believed, partly to be overlapping and partly, to some extent also to differ from the source of ROS that oxidize RNA. Although RNA and DNA oxidation are correlated, their prognostic values differ in type 2 diabetes, consistent with the view that they represent different mechanisms. The exact mechanisms need to be explored in future studies.

Between female and male subjects, large differences in excretion levels of both urinary markers were observed in this study. Although gender for long has been recognized as a con-founding factor for 8-oxodG [45], details of the physiological background of these differences are not known. It is also mainly unknown, and merit further studies, if the gender differences in levels of 8-oxodG and 8-oxoGuo are of importance for health outcomes.

In summary, this study confirms previously documented associations between 8-oxoGuo and obesity, but questions previously documented associations between 8-oxoGuo and diabetes and $HbA_{1c}$, respectively, pointing more towards a connection to hyperinsulinemia and insulin-resistance. We conclude that oxidative damage to nucleic acids, reflected in the excretion of 8-oxodG and 8-oxoGuo, gradually decreased postoperatively (except for a temporary increase in 8-oxodG immediately after surgery) and had decreased significantly two years after RYGB. This indicates that a reduction in oxidative stress could contribute to the many long-term benefits of RYGB-surgery in obesity and type 2 diabetes, including improved longevity.

## Supporting information

**S1 Fig. Research population inclusion and exclusion criteria.**
(TIF)

**S2 Fig. Individual changes in BMI and urinary markers 24 months after RYGB.** The graph is a x-y plot, where delta-values of 8-oxodG (A) and 8-oxoGuo (B) are plotted against the relative BMI for individual patients, 24 months after RYGB. On the x-axis, 0 nmol represents no change in urinary excretion of the marker. On the y-axis, 100% represents the preoperative BMI-value.
(TIF)

**S1 Table. Preoperative clinical and laboratory data for all Roux-en-Y Gastric Bypass-oper-ated patients and patients divided in subgroups according to diabetes status.** Data are reported as Mean (SD), except for the parameters reported as a percentage or number. Age is on the day of Roux-en-Y gastric bypass (RYGB) surgery. Clinical data represent the closest available before surgery. NDH, patients with biochemical glucose markers below diagnostic threshold for diabetes and not on antidiabetic treatment; DH, patients with biochemically con-firmed diabetes; DH-NDH, patients who obtained remission of diabetes after RYGB; DH-DH, patients who did not obtain remission in diabetes after RYGB; SU, Sulfonylurea; GLP-1, Glu-cagon-like peptide-1 analogue; DPP4, Dipeptidyl peptidase-4 inhibitor. *All patients also include patients for whom we were not able to confirm diabetes status. †All patients with dia-betes also include patients without hyperglycemia after RYGB, but who continued antidiabetic

medicine. ‡Apart from the HOMA-IR score and eGFR, biochemical variables are plasma or blood concentrations, unless stated otherwise with U, for urinary. Insulin, C-peptide and Glucose are fasting values. Exact *p*-values from the one-way ANOVA, followed by a Tukey post hoc test, are reported down to 0.001. Below 0.001, *P*-values are reported as < 0.001 or < 0.0001. A Welch—Satterthwaite correction followed by a Games—Howell post hoc test have been used as appropriate, to adjust for unequal variances and for HbA1c, glucose and triglycerides, significant differences were confirmed with a Kruskal-Wallis H-test, as these variables did not have a normal distribution.
(DOCX)

**S2 Table. Preoperative 8-oxodG and 8-oxoGuo, normalized to urinary creatinine.** Data are reported as mean with a 95% confidence interval. NDH, patients with biochemical glucose markers below diagnostic threshold for diabetes and not on antidiabetic treatment; DH, patients with biochemically confirmed diabetes. All patients also include patients for whom we were not able to confirm diabetes status. Exact *P*-values from unpaired t-tests are reported down to 0.001.
(DOCX)

## Acknowledgments

We would like to thank Jette Nymann and Bente Elmfeldt Madsen for assistance with biobank sample handling and analysis of cystatin C, and Katja Luntang Christensen for analysis of 8oxoGuo, 8oxodG and urinary creatinine.

## Author Contributions

**Conceptualization:** Mogens Fenger, Sten Madsbad, Henrik Enghusen Poulsen.

**Data curation:** Elin Rebecka Carlsson.

**Formal analysis:** Elin Rebecka Carlsson.

**Funding acquisition:** Sten Madsbad.

**Investigation:** Elin Rebecka Carlsson.

**Methodology:** Elin Rebecka Carlsson, Mogens Fenger, Henrik Enghusen Poulsen.

**Project administration:** Elin Rebecka Carlsson, Mogens Fenger, Henrik Enghusen Poulsen.

**Resources:** Trine Henriksen, Dorte Worm, Dorte Lindqvist Hansen, Sten Madsbad, Henrik Enghusen Poulsen.

**Supervision:** Mogens Fenger, Laura Kofoed Kjaer, Sten Madsbad, Henrik Enghusen Poulsen.

**Validation:** Trine Henriksen, Henrik Enghusen Poulsen.

**Writing – original draft:** Elin Rebecka Carlsson.

**Writing – review & editing:** Elin Rebecka Carlsson, Mogens Fenger, Trine Henriksen, Laura Kofoed Kjaer, Dorte Worm, Dorte Lindqvist Hansen, Sten Madsbad, Henrik Enghusen Poulsen.

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
