## [Decision Letter · Decision Letter 0]

22 Jun 2020

PONE-D-20-11081

Reduction of oxidative stress on DNA and RNA in obese patients after Roux-en-Y gastric bypass surgery – an observational cohort study of changes in urinary markers

PLOS ONE

Dear Dr. Carlsson,

Thank you for submitting your manuscript to PLOS ONE, which has now been thoroughly reviewed by three reviewers. After careful consideration, we feel that it has merit but does not fully meet PLOS ONE’s publication criteria as it currently stands. Therefore, we invite you to submit a revised version of the manuscript that addresses the points raised during the review process. Please submit along with the revised version of your manuscript a detailed point-by-point response to all concerns and comments. 

We look forward to receiving your revised manuscript.

Kind regards,

Yvonne Böttcher, Ph.D.

Academic Editor

PLOS ONE

Journal Requirements:

2. Please provide a sample size and power calculation in the Methods, or discuss the reasons for not performing one before study initiation.

Reviewers' comments:

Reviewer's Responses to Questions

**Comments to the Author**

1. Is the manuscript technically sound, and do the data support the conclusions?

Reviewer #1: Yes

Reviewer #2: Partly

Reviewer #3: Yes

2. Has the statistical analysis been performed appropriately and rigorously? 

Reviewer #1: Yes

Reviewer #2: Yes

Reviewer #3: No

3. Have the authors made all data underlying the findings in their manuscript fully available?

Reviewer #1: Yes

Reviewer #2: Yes

Reviewer #3: Yes

4. Is the manuscript presented in an intelligible fashion and written in standard English?

Reviewer #1: Yes

Reviewer #2: Yes

Reviewer #3: Yes

5. Review Comments to the Author

Reviewer #1: The manuscript of Carlsson et al., PONE-D-20-11081 (Reduction of oxidative stress on DNA and RNA in obese patients after Roux-en-Y gastric bypass surgery-an observational cohort study of changes in urinary markers,) describes an interesting study which deserves to be publication. However there are several shortcomings and unclear parts which should be improved.

General Remarks:

1. The description of the study is difficult to understand and a schematic illustration should be included which shows the design of the study (with exclusion ad inclusion criteria).

2. Table 1 is not clear. It is mentioned that 356 patients participated at the study but when we looked at the subgroups the overall number is only 319. Furthermore,

completely different numbers are mentioned in running text. The authors should check this very carefully.

3. It is hard to believe that blood pressure values of obese people are quite low (in the normal range) before the surgery. Did the patients receive the blood lowering medications? The authors should comment these findings.

4. It would be interesting that authors present in details data between individual weight-loss and reduction of base oxidation in form of graphs.

5. It is unclear if urinary excretion (24h) was also studied postoperatively.

6. It is in general unclear which patients were monitored and which patients dropped-out. This should be clarified in a table.

7. The authors mentioned differences between males and females (preoperative). The reason should be discussed. The authors found also a transient increase of base oxidation after the surgery, also this finding should be.

8. Discussion, Line 346: As the difference was after the adjustment not significant the authors cannot say that they confirmed early findings of an association.

9. Introduction, line 65: Further studies with patients of bariatric surgery are missing in the reference list (for example Bankoglu et al. , Mutagenesis 2018, 33, 61-67 and Bankoglu et al. Scientific Reports, 2018, 8)

Specific remarks:

Line 70: replace has by was

Line 81: correlation with weight changes

Line 136: analyses

Line 137: 310 random samples evaluated in 2015-2016 were repeated

Line 142: analyses

Line 146: 8-oxodG and 8-oxoGuo were normalized.

Line 147: The Jaffe-method assay reference is missing.

154: in this study, the reasons are explained in the section below.

Line 247: triglycerides

Reviewer #2: The manuscript investigated the impact of a weight loss in obese patients undergoing Roux-en-Y gastric bypass surgery on reduction of oxidative stress on DNA and RNA evaluating changes in urinary markers: 8-oxo-7,8-dihydro-2´-deoxyguanosine (8-oxodG) and 8-oxo-7,8-dihydroguanosine (8-oxoGuo). This work shows interesting findings, hawever I have some sugesstion:

1. the authors should briefly describe the study group

2. the authors should focus on specifically describe the patient inclusion criteria for the study, not only the exclusion criteria eg. in line 85-86 is information that ALL patients had surgery for obesity with the RYGB procedure (...); in line 87-88: Patients operated with other types of bariatric surgery, like sleeve gastrectomy or a gastric banding procedure were excluded - this this sentence is superfluous, because above the authors wrote ALL patients had RYGB

3. statistical analysis needs improvement, using a paired t Student test and calculating multiplicity adjusted p-value is more correct for 168 patients (patients who have samples before bariatric surgery and at all time intervals after surgery)

4. in the discussion, the authors describe the results generally, but nevertheless there is no information on what might have affected the results. It is known that adipose tissue is an endocrine organ that produces adipokines affects the body's metabolism. Authors should try to explain oxidative stress influence on metabolic changes in obese patients.

5. in 358: please explain the relationship between ROS formation and DNA and RNA oxidative damage

6. 8-oxo-7,8-dihydro-2´-deoxyguanosine (8-oxodG) and 8-oxo-7,8-dihydroguanosine (8-oxoGuo) are urinary markers, the authors should more explain changes in metabolic diseases

Reviewer #3: Dear Authors,

I have read your manuscript with the great interest. The experimental design is good and in my opinion the work suits the objectives of PLOS ONE. Nevertheless, I would like to address a few comments:

Introduction

The introduction should clearly state which biomarker is an indicator of DNA and which of RNA oxidative stress, what is the mechanism behind the difference. I suggest to move some information on this subject form the discussion section to introduction, so in the introduction the reader has whole background and in the discussion the authors only comment their results against results of others and do not explain the basic information. Additionally, if the order of sentences in the last paragraph was inverted, it would create better background to state aim of the study which is lacking in this version of the manuscript. If the aims are clearly stated, the material and methods section and results could be more structured and focused only on these elements that are essential for this paper. Please keep and follow the same sequence of the information in each section (Introduction (aims) – Materials and methods – Results – Discussion). Please rephrase the phrase: “longitudal changes and cross-sectional differences”. Although it may be intuitive to understand it as “long-term changes” and “differences depending on gender and heatlh status”, it should be clearly stated and explained.

Materials and Methods

The section focuses too much on exclusion criteria and therefore is too long for the purpose of this study. I understand that the author aimed to give the best possible description of the materials and methods applied, but the amount of information makes the most important information to disappear. Please provide the initial number of patients, then provide inclusion criteria and finally give the final number of analyzed cases. All additional data might be presented in supplementary materials. Also, if this is important for this study, please state why you took into account and analyzed separately the gender, the smoking status and antidiabetic treatment – it would be good to give reason for such differentiation (and how it is linked to aims of the study). A scheme of the study groups would be helpful.

Results

My major concern is connected with statistical analysis and results presentation. Distribution of normality was not checked. If a comparison of the same parameter was made at different time points, then it should be used ANOVA for the samples associated with Mauchly's sphericity test. Table 2 shows the results of the comparisons. In this situation, two-way analysis of variance with contrast analysis should be used instead of post-hoc testing.

The results are presented on the way that is very difficult to analyze it for the readers, please consider the presentation of some data in the figures, and rearrange the tables.

Please move Table 1 to supplementary materials and leave in this section only the results that are essential for the purpose of this study and linked to the aims of this paper.

Please visualize data form section about bodyweight, BMI, glucose and lipid markers after RYGB.

The quality of Figure 1 is very poor, thus I cannot really read the details. I would suggest not including the results to the Figures captions: Exact p-values are reported down to 0.001. Below 0.001, p-values are reported as <0.001 or <0.0001.

Discussion

Please add references to previous studies quoted in lines 367-369 and elaborate on it.

Summary:

The paper comprehensively analyses levels of DNA and RNA oxidative stress markers in urine samples of patients after RYGB. The authors made great effort to analyze substantial amount of data and correctly process them. In general, the paper is decently written, but it would be good to add clearly declared aims of the study and build the structure of the paper around it, and so in each section the information appears in the same order. This would make the paper more coherent. Also, the paper would benefit from moving some information to supplementary materials. Thank you for having the opportunity to review it.

Final recommendation:

Resubmit after major revision

6. PLOS authors have the option to publish the peer review history of their article (what does this mean?). If published, this will include your full peer review and any attached files.

Reviewer #1: No

Reviewer #2: No

Reviewer #3: No

---

## [Author Response · Author response to Decision Letter 0]

11 Aug 2020

Please see the attached letter "Responce to Reviewers"

---

## [Decision Letter · Decision Letter 1]

7 Oct 2020

PONE-D-20-11081R1

Reduction of oxidative stress on DNA and RNA in obese patients after Roux-en-Y gastric bypass surgery – an observational cohort study of changes in urinary markers

PLOS ONE

Dear Dr. Carlsson,

Thank you for submitting your manuscript to PLOS ONE. After careful consideration, we feel that it has merit but does not fully meet PLOS ONE’s publication criteria as it currently stands. Therefore, we invite you to submit a revised version of the manuscript that addresses the points raised during the review process.

We look forward to receiving your revised manuscript.

Kind regards,

Yvonne Böttcher, Ph.D.

Academic Editor

PLOS ONE

Reviewers' comments:

Reviewer's Responses to Questions

**Comments to the Author**

1. If the authors have adequately addressed your comments raised in a previous round of review and you feel that this manuscript is now acceptable for publication, you may indicate that here to bypass the “Comments to the Author” section, enter your conflict of interest statement in the “Confidential to Editor” section, and submit your "Accept" recommendation.

Reviewer #2: (No Response)

Reviewer #3: All comments have been addressed

2. Is the manuscript technically sound, and do the data support the conclusions?

Reviewer #2: Partly

Reviewer #3: Yes

3. Has the statistical analysis been performed appropriately and rigorously? 

Reviewer #2: I Don't Know

Reviewer #3: Yes

4. Have the authors made all data underlying the findings in their manuscript fully available?

Reviewer #2: Yes

Reviewer #3: Yes

5. Is the manuscript presented in an intelligible fashion and written in standard English?

Reviewer #2: Yes

Reviewer #3: Yes

6. Review Comments to the Author

Reviewer #2: The manuscript "Reduction of oxidative stress on DNA and RNA in obese patients after Roux-en-Y gastric bypass surgery - an observational cohort study of changes in urinary markers" is written correctly, but there are some remarks:

1.in the abstract and the main text, the term "short term" should be replaced with a specific time interval, e.g. 1 week / month after bariatric surgery

2. in line 84 the phrase "similar to sleeve gastrectomy" should be deleted

3. in line 88 and 89: "The studies investigating one or several different markers of oxidative stress and antioxidant defense after RYGB in obese patients show conflicting results (25–31)" - the sentence should be briefly expanded, what results were obtained?

4. in research population, the criteria for including / excluding patients from the study should be described clearly (comorbidities, autoimmune and viral diseases, medications taken, dietary supplements, antioxidants)

5. the conclusion should be described in more detail

Reviewer #3: Dear Authors,

I recommend the current version of the revised manuscript to be published in PLOS ONE.

Congratulations of interesting study and manuscript.

7. PLOS authors have the option to publish the peer review history of their article (what does this mean?). If published, this will include your full peer review and any attached files.

Reviewer #2: No

Reviewer #3: **Yes: **Dominika Stygar

---

## [Author Response · Author response to Decision Letter 1]

16 Oct 2020

Please see the attached letter with responce to Reviewers.

---

## [Editor Report · Decision Letter 2]

1 Dec 2020

Reduction of oxidative stress on DNA and RNA in obese patients after Roux-en-Y gastric bypass surgery – an observational cohort study of changes in urinary markers

PONE-D-20-11081R2

Dear Dr. Carlsson,

We’re pleased to inform you that your manuscript has been judged scientifically suitable for publication and will be formally accepted for publication once it meets all outstanding technical requirements.

Kind regards,

Yvonne Böttcher, Ph.D.

Academic Editor

PLOS ONE
---

## [Editor Report · Acceptance letter]

3 Dec 2020

PONE-D-20-11081R2 

Reduction of oxidative stress on DNA and RNA in obese patients after Roux-en-Y gastric bypass surgery – an observational cohort study of changes in urinary markers 

Dear Dr. Carlsson:

I'm pleased to inform you that your manuscript has been deemed suitable for publication in PLOS ONE. Congratulations! Your manuscript is now with our production department. 

Kind regards, 

on behalf of

Professor Yvonne Böttcher 

Academic Editor

PLOS ONE